# Anisotropic atom motion on a row-wise antiferromagnetic surface

Felix Zahner [1], Soumyajyoti Haldar [2] ✉, Roland Wiesendanger[1], Stefan Heinze [2,3], Kirsten von Bergmann [1] & André Kubetzka [1] ✉

Diffusion on surfaces is a fundamental process in surface science, governing nanostructure and film growth, as well as molecular self-assembly, chemical reactions and catalysis. Atom motion on non-magnetic surfaces has been studied extensively both theoretically and by real-space techniques such as field ion microscopy and scanning tunneling microscopy. For magnetic surfaces ab-initio calculations have predicted strong effects of the magnetic state onto adatom diffusion, but to date no corresponding experimental data exists. Here, we investigate different atoms on the hexagonal Mn monolayer on Re(0001) using scanning tunneling microscopy at $T$ = 4.2 K and density functional theory. Experimentally, we observe one-dimensional motion of Co, Rh, and Ir atoms on the hexagonal Mn layer, dictated by the row-wise antiferromagnetic state. Co atoms move up to 10 nm when their motion is initiated by local voltage pulses. Our calculations reveal anisotropic potential landscapes, which favor one-dimensional motion for both Rh and Co atoms, avoiding induced Rh spin moments and conserving the Co spin direction during movement, respectively. These findings demonstrate that the magnetic properties of a system can be a means to control adatom mobility, even in the case of non-magnetic adatoms.

The movement of atoms and molecules on surfaces is a fundamental and well-studied process in surface science. Adatoms typically move thermally driven by random jumps between neighboring binding sites, while longer jumps and exchange processes with the surface are activated with increasing temperature[1]. The surface symmetry plays a key role for these processes (see Fig. 1a and b): whereas diffusion is effectively isotropic on hexagonal surfaces, lower symmetry surfaces like fcc(110) can show strong directional anisotropies[2,3], or even strictly one-dimensional movement[4,5], which has been exploited for the self-organized growth of atomic chains[6,7]. For molecules, the excitation of vibrational modes can play a key role for initiating their migration or rotation[5,8]. Magnetic effects are usually neglected, either because diffusion is studied above the magnetic ordering temperature, $T_C$, or because their strengths are considered negligible compared to binding energies and diffusion barriers.

For bulk systems, however, experiments indicate that the magnetic state can affect diffusion. In bcc iron, for example, the self-diffusion coefficient systematically deviates from a linear Arrhenius relation, i.e. below $T_C$ Fe atoms diffuse more slowly through the Fe crystal than extrapolated from higher temperatures[9]. In some alloy systems, the structural composition can be influenced by applying external magnetic fields during annealing; for the antiferromagnet PdMn it has been argued that excess Pd atoms preferentially diffuse to the magnetic sublattice which is favored by the external field, thereby producing a strongly pinned magnetization, giving rise to exchange bias[10].

On magnetic surfaces, so far no diffusion experiments have been performed, but scanning tunneling microscopy (STM) manipulation experiments have reported alternating forces when a magnetic atom is moved across an antiferromagnetic surface with a magnetic tip[11,12].

[1]Institute of Nanostructure and Solid State Physics (INF), University of Hamburg, Jungiusstraße 11, 20355 Hamburg, Germany. [2]Institute of Theoretical Physics and Astrophysics, University of Kiel, Leibnizstrasse 15, 24098 Kiel, Germany. [3]Kiel Nano, Surface, and Interface Science (KiNSIS), University of Kiel, 24118 Kiel, Germany. ✉e-mail: haldar@physik.uni-kiel.de; kubetzka@physnet.uni-hamburg.de

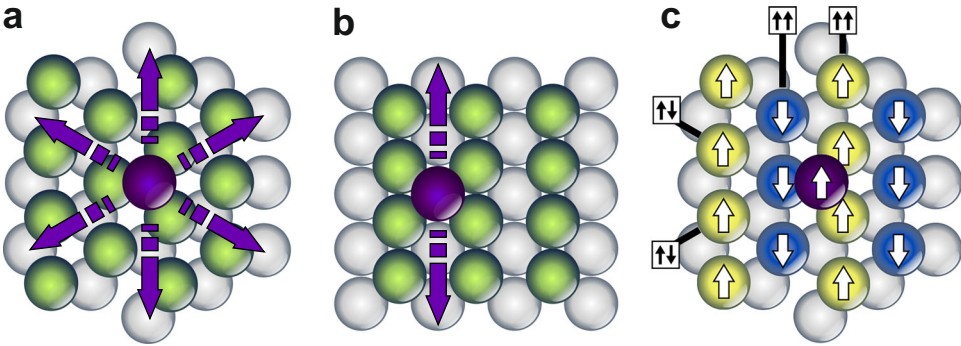

**Fig. 1 | Structural and magnetic surface symmetry. a** Quasi-isotropic diffusion of an adatom on a hexagonal surface with six structurally equivalent directions. **b** Fcc(110) surface with typically preferred diffusion along close-packed atomic rows[3,17]. **c** Adatom on a row-wise antiferromagnet, with the ↑↑-rows breaking the hexagonal surface symmetry. Its magnetic moment is assumed to be parallel to the majority of neighboring Mn surface atoms.

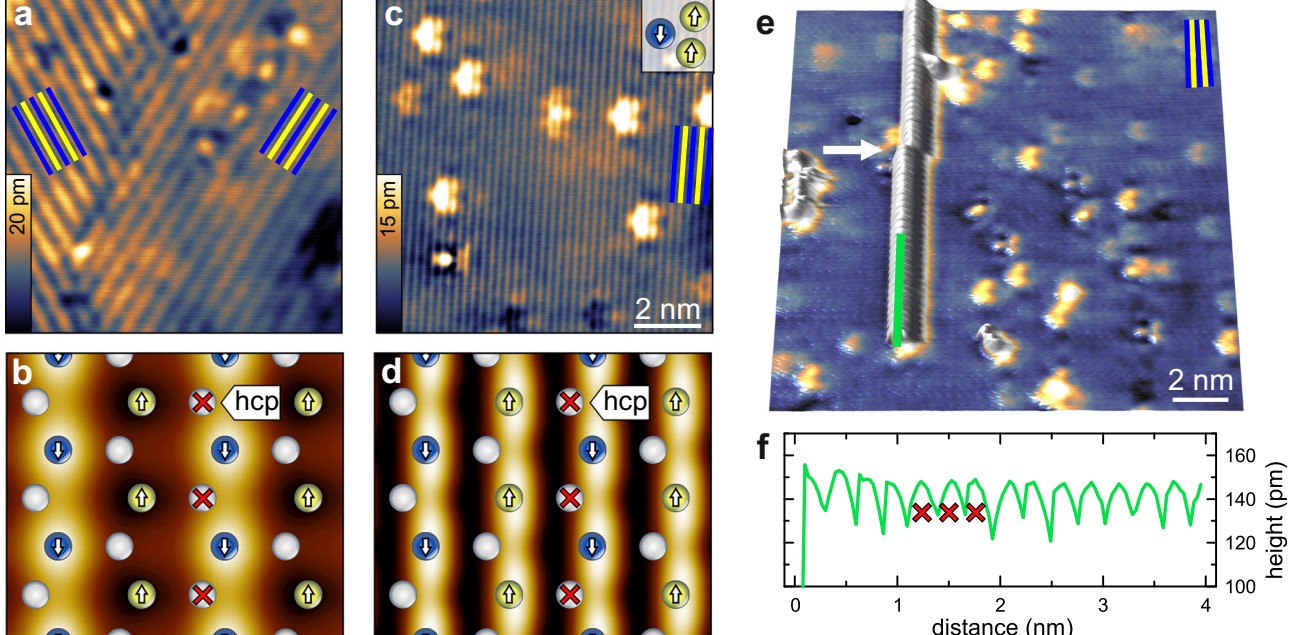

**Fig. 2 | Broken surface symmetry and one-dimensional Co atom movement.**
**a** Two rotational domains of the row-wise AFM state in fcc-Mn/Re(0001), imaged by constant current SP-STM ($U = +14$ mV, $I = 1$ nA). **b** Calculated SP-STM image at $U = +50$ mV, 3 Å above the surface based on DFT, overlaid with Mn atoms (Blue/Yellow) and surface Re atoms (Gray); adatom hcp sites are marked by red crosses. **c** High resolution (non-magnetic) STM image ($U = +30$ mV, $I = 1$ nA) of the ↑↑-rows with native defects reflecting the broken symmetry of the hollow sites, see inset. **d** DFT calculated (non-magnetic) STM image at $U = +50$ mV, 3 Å above the surface. **e** A single Co atom is imaged while it is moving to the top of the image, guided by the ↑↑-rows (slow scan direction: bottom → top, fast scan direction: left → right). **f** The line section shows that the Co atom is imaged with an atomic periodicity of 2.7 Å, see red crosses in panels **b** and **d**.

On the theory side, about 20 years ago DFT calculations for self-diffusion on Fe and Mn monolayers on W(110) and W(001) predicted a strong influence of magnetism onto adsorption sites and diffusion properties with a general trend of enhanced diffusion on ferromagnetic surfaces[13,14], but respective experiments are still missing.

Here, we investigate how different atom species move at $T = 4.2$ K on a hexagonal Mn layer hosting a row-wise antiferromagnetic (AFM) state, when their motion is initiated by high voltage scans or local voltage pulses from a stationary STM tip. For simplicity, we have chosen adatoms from the same chemical group—Co, Rh, and Ir—which are iso-electronic atoms exhibiting the same electronic shell filling. For all three atoms, we observe strictly one-dimensional movement along the rows of Mn atoms with parallel spins (↑↑-rows) (see Fig. 1c). Density functional theory (DFT) calculations show that the pseudo-morphic Mn layer is shifted laterally by 15 pm off the fcc hollow site positions on Re(0001), i.e. the symmetry of the substrate is broken by

the magnetic state electronically as well as structurally. Using the nudged elastic band method, we calculate the minimum energy paths and energy barriers for Co, Rh, and Cu adatoms and find that these differ by about 50 meV for diffusion along ↑↑-rows compared to ↑↓-rows, making their potential landscapes highly anisotropic. These DFT calculations can explain the experimentally observed one-dimensional movement of Co and Rh, however, they predict two-dimensional anisotropic diffusion for Cu adatoms.

## Results
### Experiments with adatoms
We have prepared extended Mn layers in fcc stacking on Re(0001)[15] (see Section "Experimental details" for experimental details). The hexagonal Mn layer hosts a row-wise AFM state[12,15] and in the SP-STM measurement of Fig. 2a neighboring ↑↑-rows exhibit different signal strengths[16], in agreement with the DFT calculated SP-STM image

shown in Fig. 2b, see Fig. S1 of the Supplemental Material for details. The symmetry allows for three equivalent rotational magnetic domains, two of which are seen in the spin-polarized (SP) STM image of Fig. 2a. The left domain exhibits a stronger magnetic contrast for this particular tip magnetization direction[12].

When the tunnel current is not spin-polarized, as in Fig. 2c, the magnetic state can be imaged as a stripe pattern of half the period, i.e. neighboring ↑↑-rows appear identical. This experimental data is nicely reproduced by our DFT-calculated STM image in Fig. 2d: Whereas the Mn atoms are equivalent, the ↑↑-rows are electronically different from the ↑↓-rows, due to the magnetic state. This means that the $C_3$ symmetry of the system is broken, which is also reflected by the boomerang-like shape of native defects (see Fig. 2c); the defect shape is correlated with the orientation of the magnetic state (see the "Methods" section).

With a different non-magnetic tip in Fig. 2e, the Mn layer itself appears featureless, but the direction of the ↑↑-rows can still be inferred from the defect shapes. Co adatoms on fcc-Mn/Re(0001) are easily moved during standard STM imaging: in Fig. 2e a single Co atom jumps along the ↑↑-rows and is imaged again in every line and therefore appears like an atomic chain. Only once does the atom jump one atomic row to the right (see arrow in Fig. 2e). This data is reminiscent of the experiments performed by Li et al. with Ag atoms on Ag(110)[3,17] and is a first hint that the magnetic state dictates the movement direction of Co atoms. The line section in Fig. 2f shows the period of the atomic lattice, indicating that only one type of hollow site is a stable position for Co, in agreement with no zigzag movement being observed. On all other positions, the residence time is too short to be detectable in this measurement. We will see in section "First-principles calculations" that the Co atoms prefer the so-called hcp hollow site above the surface Re atom, see red crosses in Fig. 2b and d.

To minimize the influence of the tip, instead of dragging the adatom along during imaging, as in Fig. 2e, in the following, we "kick" the adatoms with a voltage pulse from a stationary tip and determine the new position by subsequent imaging. The kicking is performed by a sudden voltage increase at a constant tip height above the atom, which also leads to an increased tunnel current (see inset of Fig. 3a and "Methods" section). By performing these experiments with one and the same micro-tip on different rotational domains, we can exclude a possible tip asymmetry being responsible for the movement direction. Figure 3a shows a surface area with two rotational domains like in Fig. 2a, this time imaged with vanishing spin contrast. The domain wall (DW), now visible via a reduced electron density[12], is marked by a white line. We find that the Co movement initiated by kicking is always toward the upper left on the left domain and toward the top of the image on the right domain, in both cases following the ↑↑-rows of the magnetic state (Supplementary Movie 1). Examples are given in Fig. 3b and c (left domain) and Fig. 3d and e (right domain). Which of the two equivalent directions along the ↑↑-rows of a particular domain is taken, can depend on the specific micro-tip: for some tips, like the one used in Figure 3a–e, one movement direction is favored; other tips show a more symmetric distribution. Independent of the micro-tip used for kicking, we always observe a strictly one-dimensional movement of Co atoms along the ↑↑-rows, within an accuracy of one atomic lattice site. Occasionally, Co atoms that are further away from the directly kicked atom also move 1–2 lattice sites (see Fig. 3b–d), indicating a weak long-range effect. These short jumps also occur exclusively along the ↑↑-rows.

The Co atoms are surprisingly mobile on fcc-Mn/Re(0001). Depending on the micro-tip, we find a threshold voltage for single-site jumps at a rather low voltage, in the range of 8–50 mV at $I = 1$ nA. In Fig. 3b–e Co atoms move 10–15 lattice sites when kicked by voltage pulses of $U_P = +200$ mV. A further measurement series of 19 kicking events, using both voltage polarities (see also Supplemental Material) is shown in Fig. 3f: the average travel distance is similar as before and

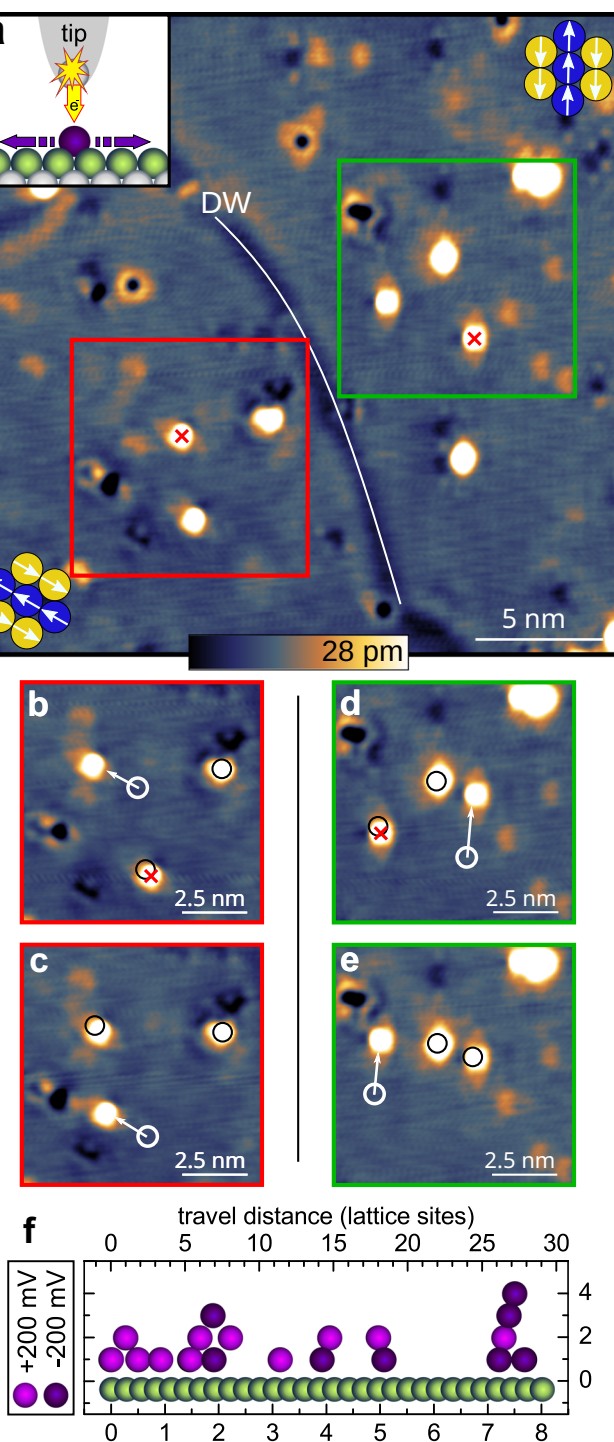

**Fig. 3 | Kicking Co atoms on a row-wise antiferromagnet. a** Overview STM image of Co atoms on fcc-Mn/Re(0001) with two rotational domains; their orientation is apparent by the defect shapes and the domain wall (DW) is imaged via a reduced electron density. Kicking is performed by a sudden voltage raise at constant tip height, see inset. **b** The upper left Co atom has moved along the ↑↑-rows after kicking. **c** The bottom Co atom has moved in the same direction after kicking. **d** The bottom Co atom has moved along the ↑↑-rows after kicking. **e** The left Co atom has moved in the same direction after kicking. (All images $U = +8$ mV, $I = 1$ nA, all voltage pulses: $U_P = +200$ mV, $\Delta t = 1$ s, tip kicking locations are marked by crosses in the previous image, previous atom positions are marked by circles.) **f** Kicking distances for both voltage polarities (color-coded), $|U_P| = 200$ mV, $\Delta t = 0.5$ s.

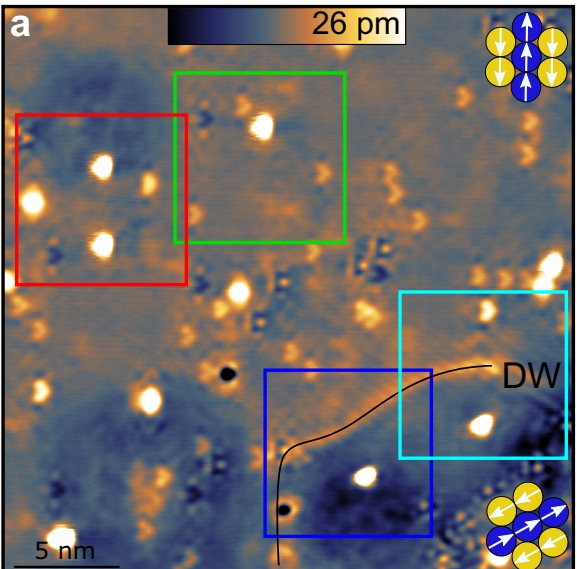
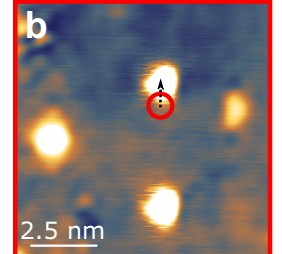
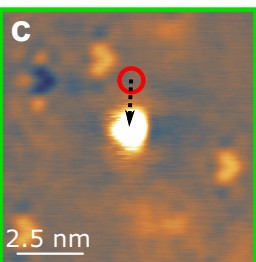
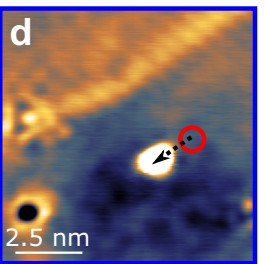
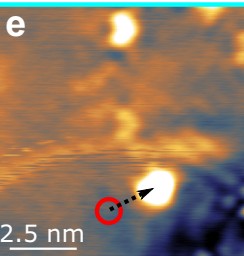

**Fig. 4 | Kicking Rh atoms on a row-wise antiferromagnet. a** Overview STM image of Rh atoms on fcc-Mn/Re(0001) with two rotational domains. A domain wall (DW) is marked by a black line. **b–e** When kicked, the Rh atoms also move according to the rotational domain state, but travel shorter distances compared to Co, despite the use of pulses with higher voltage magnitude. The electron density of native defects and also the Rh atoms are asymmetric due to the AFM state (all images $U = +25$ mV, $I = 2$ nA, $U_P = +1$ V, $\Delta t = 400$ ms).

the distribution is roughly uniform with an accumulation at the far end at 7–8 nm. We observe a maximum kicking distance of about 10 nm at $U_P = 500$ mV. A random walk-type motion of uncorrelated short jumps would lead to a normal distribution with a maximum at zero and an exponential tail[1], and can therefore be ruled out. Instead, these results indicate low diffusion barriers along the ↑↑-rows and effectively low damping, allowing the Co atoms to perform long jumps[18,19].

To make a more general case we repeated the above experiments with a second adatom species: Rh is in the same chemical group and thus iso-electronic to Co, but almost 75% heavier and typically non-magnetic on metal surfaces. We again use a surface area with two rotational domains to exclude tip artifacts (see Fig. 4a). We similarly find that the movement is always along the ↑↑-rows (Supplementary Movie 2), as can be seen in Fig. 4b, c (upper domain) and in Fig. 4d, e (lower domain). The Rh atoms show no spherical symmetry but instead reflect the broken $C_3$ symmetry, like many of the native defects. In comparison, the Co shows an asymmetry in the extended electron density surrounding the atom (see Fig. 3a–e). Apparently, Rh is much less mobile than Co: at $U_P = +200$ mV Rh atoms do not move more than one atomic site and at $U_P = +1$ V they travel only 1.0–2.5 nm (see Fig. 4b–e). We observe a maximum travel distance of 3 nm at $U_P = +2$ V.

Attempts to kick Ir atoms with similar parameters failed, indicating that Ir is even less mobile than Rh. However, in one case we were able to initiate one-dimensional movement along the ↑↑-rows of the AFM state, by imaging at high voltages and high tunnel current (see Fig. S2 of the Supplemental Material). We thus find for Co and Rh strictly one-dimensional movement, within an accuracy of one atomic site, and for Ir some indication of the same behavior. The critical parameter to move an atom seems to be the voltage magnitude $|U_P|$. The threshold voltage is tip-dependent and element-specific with $U_P = 8$–50 mV for Co and $U_P = 200$–300 mV for Rh.

**First-principles calculations**

In order to explain the experimental observations we performed DFT calculations. We find that for the fcc-stacked Mn monolayer on Re(0001) the row-wise AFM state is energetically much more favorable (by 300 meV/Mn atom) than the ferromagnetic (FM) state. The experimental result of Fig. 2c highlights the symmetry breaking of the substrate due to the row-wise AFM state. Consequently, in the

structural relaxations of fcc-Mn/Re(0001) we have also allowed for a lateral relaxation of the Mn atoms. Indeed, in addition to a relaxed Mn–Re interlayer distance, we also find a lateral shift of the Mn monolayer by ~15 pm, moving the Mn atoms of the ↑↑-rows closer to bridge sites of the substrate (see Fig. 2b and d). As a result, the induced magnetic moments of the Re surface atoms are enhanced, and the antiferromagnetic Mn–Re interaction becomes stronger (The lateral shift of the Mn layer results in a total energy gain of 18 meV/Mn atom for the row-wise AFM state. Thereby, it becomes the magnetic ground state of the system since it is lower than all spin spiral states as well as the triple-Q state[15].). Note, that an even larger magnetism-driven lateral shift occurs for an Mn bilayer on Ir(111) with a row-wise AFM state as recently reported in ref. 20. For our laterally shifted lowest energy row-wise AFM state we find that the Mn magnetic moments of $m_{Mn} \approx 3.4\mu_B$ point in-plane along the ↑↑-rows (Fig. 5a) due to the easy-plane magnetocrystalline anisotropy and the anisotropic symmetric exchange interaction[15].

To investigate the experimentally observed anisotropic motion of Co and Rh adatoms on this substrate we calculate via DFT the minimum energy paths and the related energy barriers for different directions of adatom movement using the climbing image nudged elastic band (NEB) method[21,22] (see the "Methods" section). For both atom species the preferred adsorption site is the hcp-hollow site, and we move the adatom from the initial hcp-hollow site (I) to a final adjacent hcp-hollow site (F) via different paths (see Fig. 5a). We start our discussion with the Co adatom, which possesses an intrinsic magnetic moment of about $+1.6\mu_B$ and favors a parallel alignment to the Mn atoms of the substrate, see the indicated spin in the initial state of Fig. 5a. For path 1 (blue) the adatom spin points along the same direction for states I1 and F1, whereas on path 2 (red) the adatom crosses an ↑↑-row and a spin-flip is required from I2 to F2 (see Fig. 5a). We find that on both paths the adatom roughly moves via the intermediate fcc-hollow site. Our calculations for the Co adatom (Fig. 5b) show that path 1 is symmetric, with two saddle points near the ↑↓-bridge sites, resulting in an energy barrier height of ~150 meV, and one intermediate energy minimum near the fcc hollow site. Interestingly the Co adatom does not move to the center of the fcc hollow site, where an inverted moment would be expected. Instead, the magnetic moment direction does not change along the entire path 1 and only

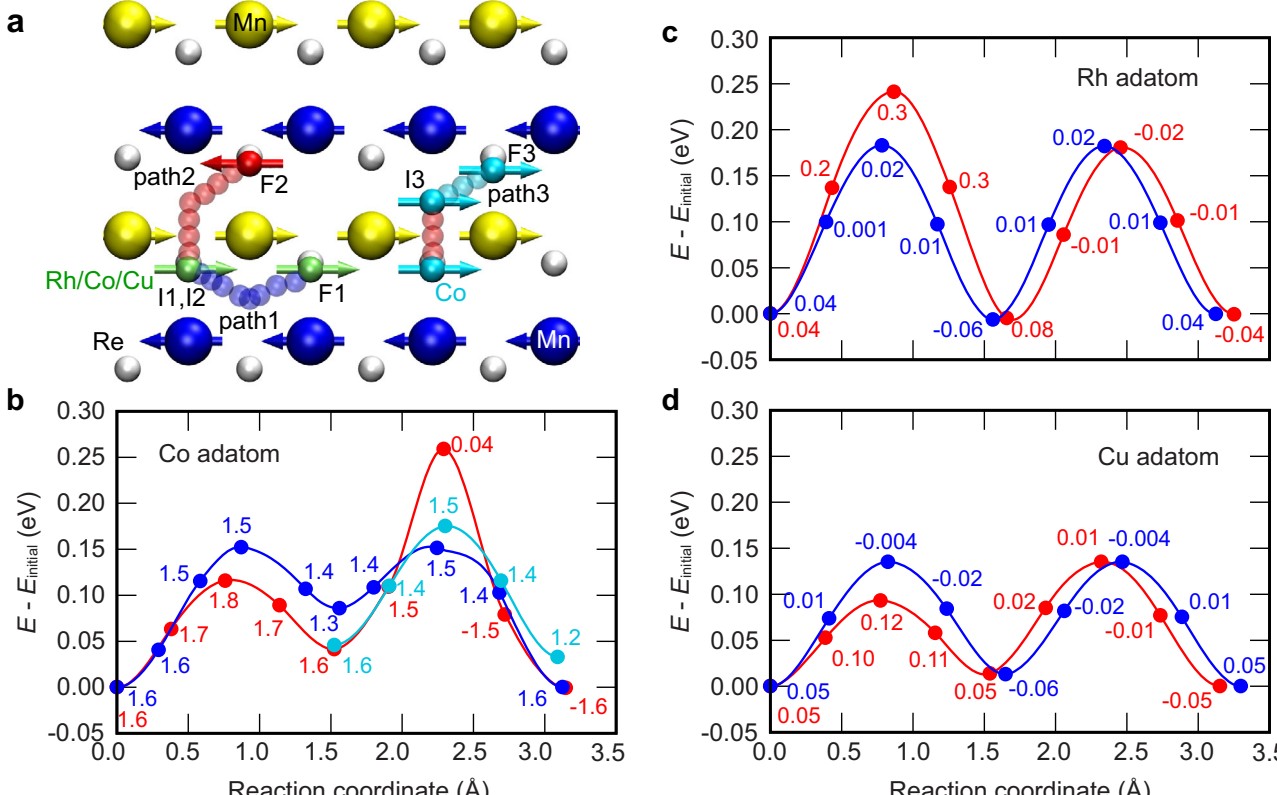

**Fig. 5 | DFT minimum energy paths and energy barriers for Rh, Co and Cu atoms on Mn/Re(0001). a** Minimum energy path for the quasi-one-dimensional movement of Rh and Co adatoms between close-packed atomic rows of the fcc-Mn/Re(0001) surface in the row-wise AFM state obtained via DFT nudged-elastic band calculations. Yellow and blue spheres with arrows represent Mn atoms with opposite spin directions. Gray spheres denote Re surface atoms, while small green, red, and cyan spheres represent adatoms. Note that the lateral shift of the Mn layer with respect to the Re surface obtained from DFT is included in the sketch. Blue and red transparent spheres indicate the DFT calculated minimum energy paths 1 and 2, respectively, from the hcp hollow adsorption site (initial states I1 and I2) to two similar adsorption sites with opposite spin directions (final states F1 and F2). For the Co atom, the minimum energy path is also given from the initial state I3 to the final state F3 (cyan spheres, path 3). **b–d** Energy along the minimum energy paths 1 and 2 for Rh, Co, and Cu atoms from the initial state I1 and I2 to the final states F1 and F2 along the red and blue paths indicated in panel **a**, respectively. For the Co atom, the energy is also given along minimum energy path 3 from I3 to F3 (cyan symbols and line). The magnetic moment of the Rh, Co, and Cu atom is given in $\mu_B$ at every image along the paths.

shows slight variations of its magnitude, as indicated by the numbers in Fig. 5b.

On path 2 (red) the Co adatom crosses two inequivalent bridge sites and the preferred spin directions in I2 and F2 are opposite, resulting in a more complex minimum energy path. We find that the saddle point to cross the ↑↑-bridge site of path 2 has lower energy compared to the ↑↓-bridge sites of path 1. In contrast to path 1, on path 2 the Co crosses the center of the fcc-hollow site, as the respective spin direction is favored in this case, compare intermediate minima of paths 1 and 2. The second part of path 2 exhibits a high energy barrier of ~260 meV at the ↑↓-bridge site, where the Co magnetic moment is nearly quenched ($0.04\mu_B$), which results from the required spin-flip along the path and the constraint to collinear magnetic moments in our calculation[14]. Finally, the Co adatom arrives at F2 where it has the opposite spin direction compared to I2. With an overall barrier height of ~260 meV, path 2 is clearly unfavorable with respect to a Co movement along ↑↑-rows (path 1), consistent with our experimental observations.

To reveal the role of the Co spin–flip on the energy barrier for the path across an ↑↑-row, we have performed a minimum energy path calculation from I3 to F3 with the constraint that the Co adatom spin cannot flip (cyan spheres and line in Fig. 5a and b). We find that now the energy barrier is significantly reduced for the ↑↓-bridge site, with a Co magnetic moment of $1.5\mu_B$. The spin of the Co atom may flip after the metastable state F3 has been reached such that relaxation into the favorable state F2 is achieved. The initial state I3 of path 3 coincides

with the intermediate energy minimum of path 2 (see Fig. 5a and b). Therefore, we can combine the first half of the path 1 with path 3. While this new path 3 has a significantly lower energy barrier compared to path 2, where the spin flip occurred near the bridge site, the lowest energy path for the Co adatom remains path 1, where the Co adatom effectively moves parallel to the ↑↑-row, in agreement with the experiment.

Next, we study the movement of a Rh adatom along paths 1 and 2 (see also Supplementary Fig. 3). The induced magnetic moment in the hollow sites is on the order of only $0.04$–$0.08\mu_B$, (see Fig. 5c). The ↑↓-bridge site energy barriers of the symmetric path 1 are lower than the ↑↑-bridge site on path 2. The different magnetic moments of the Rh adatom in the different bridge sites, i.e. $0.02\mu_B$ versus $0.3\mu_B$ at ↑↓- and ↑↑-bridge sites, respectively, suggest that a high induced moment is unfavorable and leads to the increased energy barrier. In contrast to the case of the Co adatom, where the reduction of the magnetic moment leads to an increase of the energy barrier, in the case of Rh a large magnetic moment results in an increased barrier. Based on the local density of states (LDOS, see Supplementary Fig. 4), we attribute this effect to the spin-dependent hybridization of the adatoms with the Mn surface atoms which is distinctively different for Rh and Co. For the Rh atom, the increased spin moment at the ↑↑-bridge site is connected to an energetically unfavorable enhancement of the LDOS close to the Fermi energy. Despite this opposite behavior, Co and Rh adatoms both have path 1 as the lowest energy path, as also seen in the experiments of Figs. 3 and 4. Note, that for the Rh atom motion, the magnetism-

induced lateral shift observed for the Mn layer on Re(0001) plays a key role. NEB calculations for an unshifted Mn layer lead to almost isotropic minimum energy paths (see Supplementary Fig. 5).

To facilitate a more complete understanding of magnetism-induced anisotropic motion we also discuss our result for a Cu adatom (see Fig. 5d). The induced magnetic moment of the Cu adatom is on a similar scale as that of the Rh atom. However, in contrast to the case of Rh, an increased Cu magnetic moment at the ↑↑-bridge site is accompanied by a lower energy barrier on path 2 as compared to the energy barrier at the ↑↓-bridge along path 1. Due to its fully occupied $3d$ bands, Cu exhibits only a small LDOS at the Fermi energy. Therefore, the hybridization occurs dominantly between the occupied $3d$-bands of Cu and Mn which can explain the different behavior (Supplementary Fig. 4). In consequence, the predicted motion of the Cu adatom is still anisotropic, but in contrast to Co and Rh the preference is along the two ↑↓-row directions of the row-wise AFM state, allowing for two-dimensional diffusion.

## Discussion

The experimentally found higher mobility, i.e. longer travel distances of Co atoms, agrees qualitatively with a lower barrier of -150 meV compared to -180 meV for Rh. Additional effects beyond this adiabatic analysis might contribute to the high mobility of Co. Firstly, our DFT calculations show a total binding energy of Co atoms on Mn/Re(0001) which is 1.5 eV lower than for Rh atoms. This means that a ballistic movement at an increased distance from the surface[23] requires less energy for Co atoms than for Rh atoms. Since in the experiment some of the energy of the tunneling electrons is used for the excitation, the excess energy could be larger for the Co atoms. This is in accordance with the larger threshold voltage observed for Rh atoms. Secondly, the $3d$ orbitals of Co have a smaller extent than the $4d$ orbitals of Rh, which may result in a reduced interaction with the surface and a reduced damping for a moving Co atom. Thirdly, Rh is about 75% heavier. The lighter and faster moving Co atom can therefore cover more distance before it comes to rest. This argument only holds for damping processes that scale with travel time[24]. Finally, the spin degree of freedom might contribute to the excitation process and the travel distance. For molecules, threshold voltages were measured via action spectroscopy, and vibrational modes were identified to play a key role in initiating molecular movements[5,8]. Similar experiments might shed light on the excitation process for adatoms on a magnetic surface. In addition, long jumps have been found experimentally for molecules[25,26] and one reason is their ability to store energy during movement in the form of vibrations. Spin excitations of the Co atom or within the Mn layer might play a similar role here.

Overall, the first-principles calculations demonstrate that the experimentally observed anisotropic motion of adatoms originates from the anisotropic potential landscape of the row-wise AFM state. The spontaneous magnetic ordering is accompanied by an electronic anisotropy and a magnetism-induced lateral shift of the Mn layer relative to the substrate. In contrast to diffusion on surfaces with low structural symmetry such as Ag(110)[17] or Cu(211)[27], where all adatoms are expected to be guided by the geometric anisotropy, in the case of magnetic symmetry breaking other competing effects come into play: the preference of an adatom for maximizing or minimizing its (induced) magnetic moment has an impact on the height of energy barriers, and the spin-dependent hybridization of the adatom with the anisotropic electronic states of the magnetic surface cannot be neglected for the favored direction of motion. This anisotropic motion provides several possibilities for control: antiferromagnetic domains can be switched by lateral currents[28], which in principle allows for external control of diffusion directions on antiferromagnetic surfaces; different adatoms can show different preferred directions of motion, as predicted for instance for Co and Cu. In conclusion, we have demonstrated that the magnetic properties of a surface can play a

decisive role in controlling atomic motion, with possible consequences for related phenomena such as nanostructure growth, molecular self-assembly, and catalysis.

## Methods

### Experimental details

The experiments were conducted in an ultra-high vacuum system with different chambers for substrate cleaning, film growth, and STM measurements. The Re(0001) single crystal surface was cleaned by repeated heating cycles in an oxygen atmosphere of $10^{-7}$–$10^{-8}$ mbar at temperatures of up to $T = 1400$ K; before metal deposition, a final flash to $T = 1800$ K was performed. The Mn was evaporated from a pyrolytic boron nitride (PBN) Knudsen cell of volume 2 cm³ held at $T = 670$ °C, at a rate of about 0.1 atomic layers per minute with the Re single crystal at an elevated temperature (≈100 °C) from the final flash.

The Mn/Re(0001) samples were then cooled to $T = 4.2$ K within the STM. After a quick transfer to the e-beam evaporator ($\Delta t < 5$ s) Co or Rh atoms were evaporated from a rod of 2 and 1 mm diameter, respectively, at a rate of about 0.1 atomic layers per minute. We estimate a surface temperature of $T = 25 \pm 5$ K during adatom deposition.

We use a Cr bulk STM tip, etched in 1 M HCl solution. Within the STM the tip was cleaned by field emission on a W(110) surface and sharpened by voltage pulses of $U = 4$–6 V as well as gentle collisions with the Mn/Re(0001) surface. Except for the measurement shown in Fig. 2a we did not optimize the tip for magnetic contrast. Consequently, in all other data, the tip exhibits no significant spin sensitivity at the used tunnel voltages, thereby simplifying the data analysis. Magnetic domain walls were imaged by making use of their (spin-averaged) electronic signal[12] and the orientation of rotational domains was determined via the symmetry of native defects: Supplementary Fig. 6 shows a surface area with all three rotational domains, demonstrating the strict correlation of domain orientation and defect shape orientation.

For atom "kicking", the Co or Rh atom was first moved to a defect-free area and the tip was then stabilized precisely above the atom, using an atom-tracking feedback by steepest ascent (see https://www.specs-group.com/nc/nanonis/products/detail/atom-tracking/ for details) and tunnel parameters on the order of $U = +10$ mV and $I = 200$ pA. The atom-tracking and height-control feedback were then switched off and the voltage raised abruptly at constant tip height for $\Delta t = 0.4$–1.0 s to $U_\text{P} = \pm 200$ mV and $U_\text{P} = +1$ V for Co and Rh atoms, respectively. A voltage pulse is thus automatically accompanied by an increased current through the adatom, but only as long as the atom does not move. For the measurement series displayed in Fig. 3f, the tip was stabilized above the atoms at $U = +10$ mV and $I = 200$ pA, and after the kick the atoms were moved back to their original positions using constant current pulling mode at approximately $U = 4$ mV and $I = 60$ nA. Since micro-tip changes are more likely during manipulation than during imaging, the necessity to move an atom back into position, due to surrounding defects, limits the number of kicks which can be performed with the same micro-tip. The data of Fig. 3 has been smoothed by a 3-pixel Gauss filter; all other figures show raw data.

### Computational details

We have carried out first-principles calculations based on density-functional theory (DFT) using the projector-augmented wave (PAW)[29] method which is implemented in the VASP code[30,31]. We have performed spin-polarized calculations using Wigner−Seitz radii for the elements, i.e. $R_\text{Mn}^\text{WS} = 1.32$ Å, for Mn and $R_\text{Re}^\text{WS} = 1.43$ Å for Re. As lattice parameters, we used $a_\text{NN} = 2.78$ Å, $c = 4.49$ Å, which were taken from ref. 32 obtained via DFT within the generalized gradient approximation (GGA). We have designed the films with fcc-stacking of the Mn monolayer using an asymmetric film consisting of 6 Re(0001) layers with an Mn layer on top. Structural relaxations of this film have been performed within the row-wise AFM state of the Mn layer and using the

PBE exchange-correlation (xc) potential[33]. We have used a $(5 \times 6)$ supercell in order to design the row-wise AFM state. The large supercell was needed to avoid the periodic image interaction of the adatoms. The Mn layer and the uppermost Re layer were relaxed in the $x$, $y$, and $z$-direction. The interlayer distances of the bottom 5 layers of Re were kept constant at the bulk reference value of $c/2 = 2.24$ Å. We used $5 \times 5 \times 1k$ points in the irreducible wedge of the two-dimensional (2D) Brillouin zone (BZ) and a plane wave basis set cutoff of $E_{max} = 350$ eV. The structures were optimized using the conjugate gradient method with forces calculated from the Hellman–Feynman theorem. Structures were considered to have been optimized when all the forces were smaller than 0.01 eV/Å. To calculate the most favorable position for the adatom adsorption, we have used the optimized Mn/Re(0001) surface structure from the above and placed the adatom (Co, Rh, and Cu) at different sites. Here we have only allowed the adatom to relax in $x$, $y$, and $z$-direction. For all calculations with the Co, Rh, and Cu adatoms we have used the same parameters and the same PBE exchange-correlation potential as for the relaxation of the Mn/Re(0001) surface.

The Co atom preferentially adsorbs in the hcp hollow site (indicated by I1 in Fig. 5a), which is about 50 meV lower in energy compared to the fcc hollow site. In both cases, a small lateral shift relative to the center of the Mn atom triangle is found. The configuration of the three nearest Mn magnetic moments results in a net ferromagnetic coupling to the Co magnetic moment ($1.6\mu_B$), where two of them are parallel and one is antiparallel to the Co moment (Fig. 5a). For the Rh atom the two hollow sites are almost energetically degenerate, with a difference of only ~4 meV, and in the hcp site, the adjacent Mn atoms induce a Rh magnetic moment of ~$-0.04\mu_B$.

The magnetic moment of the Co adatom at hcp hollow site F2 (Fig. 5a) is $-1.6\mu_B$ due to the magnetic moment direction of the two nearest-neighbor Mn moments (net ferromagnetic coupling). In order to obtain a net antiferromagnetic coupling to the Co magnetic moment with the three nearest-neighbor Mn atoms, we perturbed the position of the Co adatom by a small amount (-0.05 Å) and relaxed the structure with an initial high value of positive magnetic moment for the Co adatom (net antiferromagnetic coupling with the three nearest neighbor Mn atoms). Upon structural relaxation, we find that the Co atom moved to position F3 which is shifted by 15 pm with respect to F2 in the direction perpendicular to the ↑↑-rows. At site F3 the magnetic moment of the Co atom is $+1.2\mu_B$ with a net antiferromagnetic coupling to the three nearest neighbor Mn atoms. Our calculation shows that F3 is a higher energy local minima state by ~40 meV compared to the F2 state with an opposite spin moment.

We calculate the energy barriers and the minimum energy paths between two known adatom adsorption sites by using the climbing image nudged elastic band method (NEB)[21,22]. In the NEB method, we optimize a number of intermediate images to their lowest energy possible while maintaining equal spacing to neighboring images. This is obtained by adding a spring force along the band between the images. The climbing image method is a small modification to the NEB method where the highest energy image is pushed up to the saddle point to get the exact position of the saddle point. This is done by inverting the true force for this image along the tangent. Thus this image does not feel the spring force. In our NEB calculations, we have not relaxed the surface atoms for simplicity. We have used a mixture of quick-min and LBFGS optimizers[34] and $-5.0$ eV/Å$^2$ spring force for our calculations.

For the simulated STM and SP-STM images of Mn/Re(0001), we have used the full-potential linearized augmented plane wave method (FLAPW) as implemented in the FLEUR code (see https://www.flapw.de). We have used the optimized Mn/Re(0001) surface structure obtained from the VASP code and performed a spin-polarized calculation. In all FLEUR calculations, we have used the Wigner–Seitz radii for the respective element used in the VASP calculations as the radii of the muffin tin spheres. Here we have used $1444k$ points in the full

Brillouin zone and an energy cutoff of 4.3 a.u.$^{-1}$ to obtain a good description of the spin-dependent local density of states in the vacuum region. Note that the FLEUR code was used for the simulation of STM and SP-STM images since it provides a very accurate description of the vacuum region. Since the film geometry is implemented in the FLEUR code, the exponential decay of the wave functions and of the charge and magnetization densities in the vacuum above the surface is well described by the choice of the basis functions in this region which also exhibit an exponential decay[35,36].

## Data availability
The STM and DFT data are available upon reasonable request.

## Code availability
The STM data was analyzed with the open-access software Gwyddion (http://gwyddion.net).

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

## Acknowledgements

We acknowledge funding from the Deutsche Forschungsgemeinschaft (DFG, German Research Foundation) under project numbers 408119516, 418425860 and 445697818 and financial support from the Open Access Publication Fund of Universität Hamburg. This work was performed using HPC resources from the North-German Supercomputing Alliance (HLRN).

## Author contributions

A.K. devised the experiments. A.K. and F.Z. prepared the samples, performed the STM experiments and discussed the results with K.v.B. and R.W. Calculations were performed by S.H. and analyzed together with St.H. The figures were prepared by A.K., F.Z. and S.H. The manuscript was written by A.K., K.v.B. and St.H. All authors discussed the results and contributed to the manuscript.

## Funding

## Competing interests

The authors declare no competing interests.
