## [Transparent Peer Review file · Nature Communications]

Anisotropic atom motion on a row-wise antiferromagnetic surface

Corresponding Author: Dr André Kubetzka

Version 0:

Reviewer comments:

Reviewer #1

(Remarks to the Author)

The authors reported the effect of a magnetic substrate on the diffusion of both magnetic (Co) and non-magnetic (Rh) atoms. The results are well described with good quality data and the conclusions are supported by theoretical calculations. The article presents novelty and can be of interest for specialist in the field, however the topic is too specific with restricted impact. It is not clearly demonstrated that the work has the potential necessary for publication in Nature Communications. It would be better suitable for a different journal.

Minor comments:

1. In the first paragraph of the introduction the authors state: "whereas diffusion is effectively isotropic on hexagonal surfaces"; the process is not totally isotropic since there are six preferred diffusion directions as correctly state in figure 1a caption.
2. A short description of the experiment/sample preparation should be added in the last paragraph of the introduction or the beginning of the experimental section.
3. The discussion mentions additional calculations for Cu atoms, a supporting figure should be added with these results.

Reviewer #2

(Remarks to the Author)

Please see attachment

Reviewer #3

(Remarks to the Author)

In this work, F. Zahner and co-workers observed the one-dimensional movement of both magnetic Co atoms and non-magnetic Rh atoms on the surface of AFM ordered Mn layer. Without the inclusion of first-principle calculations, this observation seems to contradicts the conclusion that 'magnetism can play a decisive in controlling atom movement', which is drawn at the end of the article. Even though first-principle calculations show the one-dimensional motion of Co and Rh atoms could result from different kinds of interactions with the magnetic substrate, this serves as weak evidence. Although the conclusion is not surprising, I found the evidence insufficient and the discussion unpersuasive. Therefore, I cannot recommend the publication of this work in Nature Communications. Below are specific issues that need to be addressed:

1. If the author observed the one-dimensional movement of magnetic Co atoms and free movement of non-magnetic Rh atoms, it directly indicates the decisive role of magnetism in controlling atom movement. However, the observation of the one-dimensional movement of both magnetic Co atoms and non-magnetic Rh atoms makes the role of magnetism questionable.
2. A control experiment above the Neel temperature of Mn layer will ultimately resolve the debate. Without additional experiments, a comprehensive discussion to exclude alternative explanation, such as the influence of anisotropic atomic structure shown in Figure 2d and Extended data Figure 3c, is necessary before concluding the decisive role of magnetism.
3. Figure 2a-d shows the anisotropic feature in both SP-STM images and normal STM images of the Mn layer, which can be used for the identification of the spin orientation in the Mn layer. However, the description of the figure is insufficient and

therefore confusing. For example, it does not explain the origin of stripe pattern in the normal STM image (Figure 2d). It does not explain what kind of native defects it is and how they correlate with the spin orientation. It seems based on the symmetry. The discussion should be elaborated to ensure that readers have a clear understanding of these images. One suggestion for improvement is to present the simulated SP-STM images and normal STM images of the Mn layer in Figure 2. It demonstrates that the different stripe patterns are observed in SP-STM images and normal STM images, arising from AFM magnetism and structural relaxation, respectively.

4. The author use blue and yellow color in Figure 2a, b, d to indicate the spin rows. But the blue color is difficult to discern in the STM images. Please consider changing the color for better visibility.

5. When discussing the mobility of Co atoms on fcc-Mn/Re(0001), the descriptions, such as 'Co adatoms on fcc-Mn/Re(0001) are easily moved during standard STM imaging...' and 'The Co atoms are surprisingly mobile on fcc-Mn/Re(0001)', are misleading. This implies the author can distinguish the fcc-Mn/Re(0001) from the STM images and establish the correlation between high mobility of Co atom with the fcc site purely based on experimental observation. But this is not the case. In my opinion, the author should not claim atomic site until they demonstrate the different stability of adatom on fcc and hcp site through first principle calculation.

6. The discussion in section II is a bit difficult to follow. Please consider modify it to improve the clarity.

7. The conclusion section is too brief. It is not clear how the findings impact 'the related phenomena such as nanostructure growth, molecular self-assembly and catalysis. It needs more elaboration to highlight the significance of the results.

Reviewer #4

(Remarks to the Author)

Zahner and coworkers show a conclusive study of the one-dimensional movement of Co and Re atoms on a Mn thin film grown on Re(0001). STM measurements show a preferred motion along one spatial direction despite the 6-fold symmetry of the non-magnetic lattice. Hence, the authors conclude that the row-wise anti-ferromagnetic structure (AFM) breaks that symmetry and gives a one-dimensional motion. This is most clearly demonstrated by comparing the adatom motion on different rotational domains measured with exactly the same tip apex, thereby excluding many possible tip artifacts. The authors use a comprehensive set of DFT calculations to derive at this conclusion. That theoretical modeling also gives insight into the energy barriers that dominate the adatom motion.

I find these results very clear and the experiments and DFT studies are well executed. I think that it is rather surprising that nobody has studied this effect before, either in SPM or in ensemble techniques. Are the authors sure about the lack of previous studies? I am not aware of any such studies either. I strongly support publication with minor changes.

Important points:

1. Figure 3 makes it clear that there is a one-dimensional motion. However, it is not mentioned what determines upwards vs downwards motion along that axis? This is only answered at the end of the experimental methods. I strongly recommend to move that argument into the main text to not confuse the reader.

2. The threshold for the action spectroscopy is rather unspecific: 8-50mV for Co atoms. In typical action spectroscopy (Kawai group and others), these thresholds are rather precise and typically stem from inelastic excitation of vibrational excitation. Can the authors discuss this comparison in the text?

3. The methods section "B" related to DFT calculations is not clear on the following points:

a. The authors mention that "Films with fcc-stacking of the Mn monolayer were structurally relaxed using the GGA exchange-correlation potential". Were other functionals used for Co and/or Rh?

b. The authors mention that they use an "asymmetric film consisting of six Re(0001) layers". Is the same cell used in the simulation of the STM images? Did changing the code from VASP to FLEUR induce any changes in geometry/force?

c. Why was the code changed between the first and last part of the simulations? VASP does have the capability of simulating STM images (for example using the STMpw code). If the change of code was motivated by necessity or ease of performing a certain type of calculation it should be noted here. Authors should specify if the structures had to be re-relaxed or were used as-is.

d. It would be helpful to first state all commonalities of the DFT calculations (i.e. cell size, functional, k-points, cutoffs) then contrast the differences for the individual elements.

Minor points:

4. The title seems a bit unscientific "Kicking atoms"

5. Please add line numbers to make reviewing easier.

6. Page 4 top "In this nonmagnetic data" – this seems like wrong grammar.

7. Page 7 "latter saddle point" – the word latter seems wrong here

8. Page 10, line 11: the word "respectively" is not matched to anything

Reviewer #5

(Remarks to the Author)

Version 1:

Reviewer comments:

Reviewer #1

(Remarks to the Author)

The authors have addressed most of the referees comments and improved significantly the manuscript. I now recommend its publication.

Reviewer #2

(Remarks to the Author)

The authors have addressed the comments and feedback from the first review; however, I am still not convinced that this work is novel enough to be published in Nature Communications. The manuscript may be better suited for more specialized journals.

1) The authors claim that this work could be highly useful for material growth, catalysis, and rethinking the growth modes of magnetic films and nanostructures. However, the core of the work involves using an STM tip to move atoms, and while there may be some correlation between atomic motion and magnetic properties, I believe this work is only applicable to very specific cases, not as broadly as the authors suggest. Additionally, I would like to ask the authors to clarify if the Cu, Co, and Ru on the Mn layer atop the Re(0001) system have any particular material properties that should be considered in this context.

2) The authors repeatedly mention that 'kicking atom' is a new technique and that 'magnetic properties of a surface can play a decisive role in controlling atomic motion'. However, the concept of 'kicking atoms' has been previously demonstrated in numerous studies under the terms 'lateral and vertical manipulation', performed on metal, semiconductor, and insulating surfaces using electric fields, voltage pulses, Van der Waals forces, etc. There have even been efforts to build magnetic atomic chain structures through atomic manipulation. Furthermore, I still cannot find a clear explanation in the authors' reply regarding how magnetism plays a key role in atomic motion.

For these reasons, I cannot recommend this manuscript for publication in Nature Communications.

Reviewer #3

(Remarks to the Author)

I find the manuscript has improved greatly, and most of my comments have been adequately addressed. However, one major concern remains that need to be clarified before the manuscript can be considered for publication.

Geometric Anisotropy versus Magnetic Interactions:

In line 252-253, the authors mentioned that the adatoms are expected to be guided by the geometric anisotropy on surfaces with low structural symmetry, such as Ag(110) or Cu(211). A similar geometric anisotropy is present on Mn/Re(0001), as demonstrated by the prominent stripe pattern in the non-magnetic STM image shown in Figure 2c. This raises the possibility that the geometric anisotropy could also account for the observation of the one-dimensional movement of Co, Rh, and Ir atoms. Before excluding this possibility, it is challenging to conclusively determine the critical role of magnetic interaction in the atomic diffusion.

I believe the authors recognize that the predicted distinct diffusion behaviour of Cu adatoms is key to substantiating their conclusion. Unfortunately, the lack of experimental evidence leaves the diffusion behaviour of Cu atoms remains at a theoretical hypothesis stage. I strongly suggest that the authors include a direct investigation of Cu diffusion on Mn/Re(0001) surface.

Reviewer #4

(Remarks to the Author)

The authors properly addressed my concerns. I am fine to publish as is.

Reviewer #5

(Remarks to the Author)

Version 2:

Reviewer comments:

Reviewer #2

(Remarks to the Author)

I still believe there are several points of disagreement between the reviewers (#2 and #3) and the authors. However, I agree with the authors overall and therefore now recommend the manuscript for publication.

Reviewer #3

(Remarks to the Author)

The authors have addressed my questions. I recommend the publication of the current version as it is

Response to the reviewers

We thank all reviewers for their comments and suggestions, which helped improving the clarity of our revised manuscript. Upfront, we would like to clarify one issue which led to a reluctance of some of the reviewers to support our manuscript for publication in Nature Communications: the fact that Co and Rh atoms show a similar moving behavior.

For the investigated Mn layer on Re(0001), the hexagonal surface symmetry is broken within every rotational magnetic domain due to the magnetic state, both electronically (see new Fig. 2c,d) and structurally by a ~ 15 pm lateral shift of the Mn layer with respect to the Re surface. Due to the broken symmetry, even (fictional) strictly non-magnetic adatoms may move anisotropically. For a magnetic adatom, additional interactions can contribute to the diffusion behavior. For instance, the Co atom prefers the $\uparrow\uparrow$ -bridge site over the $\uparrow\downarrow$ -bridge site due to Heisenberg exchange, whereas the Rh atom prefers the $\uparrow\downarrow$ -bridge, where it can keep its low magnetic moment. The fact that both atoms show one-dimensional movement along the $\uparrow\uparrow$ -rows does not mean that magnetic interactions do not play a role, or that the surface alone can determine the movement direction for all atoms. To make this point clear, we have performed additional calculations and show our results for Cu atoms explicitly, which predict anisotropic two-dimensional diffusion, avoiding the $\uparrow\uparrow$ -row direction, in contrast to Rh and Co. The origin of the different behavior can be explained by spin-dependent hybridization of adatom and Mn surface atoms which depends on the electronic structure of the adatom as shown based on the local density of states (new Supplementary Figure 3).

The fact that atoms such as Rh and Cu, which only have a small induced magnetic moment, are affected in their motion by the AFM state is not a contradiction of our reasoning but rather widens the scope and impact of our findings that the magnetic state of a surface can play a decisive role for adatom diffusion.

Reviewer #1 (Remarks to the Author):

The authors reported the effect of a magnetic substrate on the diffusion of both magnetic (Co) and non-magnetic (Rh) atoms. The results are well described with good quality data and the conclusions are supported by theoretical calculations. The article presents novelty and can be of interest for specialist in the field, however the topic is too specific with restricted impact. It is not clearly demonstrated that the work has the potential necessary for publication in Nature Communications. It would be better suitable for a different journal.

We thank the reviewer for her/his time and for acknowledging the quality of our data and the novelty of our results. However, we respectfully disagree with the assessment of restricted impact of our work. Adatom diffusion is a core mechanism in surface science and we demonstrate for the first time that the magnetic state of a surface can play a decisive role in it. This new insight may be exploited in the future for higher level processes such as material growth and catalysis. For instance, it might be necessary to rethink growth modes of magnetic films and nanostructures, because adatom diffusion will be different, once a film becomes magnetic during growth.

Minor comments:

1. In the first paragraph of the introduction the authors state: "whereas diffusion is effectively isotropic on hexagonal surfaces"; the process is not totally isotropic since there are six preferred diffusion directions as correctly state in figure 1a caption.

Indeed, a hexagonal surface is not "totally isotropic", therefore we have used the term "effectively isotropic" in contrast to highly anisotropic surfaces like fcc(110).

2. A short description of the experiment/sample preparation should be added in the last paragraph of the introduction or the beginning of the experimental section.

We thank the reviewer for the suggestion. Accordingly, we added a sentence at the beginning of the experimental part, where we refer to the methods sections for details.

3. The discussion mentions additional calculations for Cu atoms, a supporting figure should be added with these results.

We now show the additional calculations for Cu as part of the new Fig. 5. Copper is an important addition, as the calculated preferred diffusion direction is different to those of Co and Rh, which demonstrates that the surface alone cannot determine the diffusion properties. We have also included the minimum energy paths and spin-resolved local density of states for all three atom types in the Supplementary Information (new Supplementary Figures 2 and 3) which shows their difference in terms of the spin-dependent hybridization between adatom and Mn surface atoms. We hope that with these clarifications the reviewer can support publication of our revised manuscript in Nature Communications.

Reviewer #2 (Remarks to the Author):

(text inserted from the pdf attachment)

F. Zahner et al., have reported STM and DFT studies on lateral motion of Co and Rh atoms induced by voltage pulses from a STM tip on magnetic surfaces. The authors have measured and simulated that the atoms' preferred movement is associated with magnetic properties of the surfaces. I believe that the manuscript is in general written well. However, I do not think that this work meets the high standard of novelty required by Nature Communications for the following reasons.

We thank the reviewer for her/his time and efforts. The presumed lack of novelty is unsubstantiated and at odds with the assessments of reviewer #1 and #4. Our view is that we have employed a *new* technique (kicking atoms) to investigate a *new* so far overlooked effect. Or, as Reviewer #4 puts it: "I think that it is rather surprising that nobody has studied this effect before, either in SPM or in ensemble techniques."

1) In the introduction section (pg 2-3), the authors mentioned that "For bulk systems, however, experiments indicate that the magnetic states can affect diffusion" and "On magnetic surfaces, so far no diffusion experiments have been performed". This sounds to me that the diffusion direction of the adatoms could be controlled by the magnetic interaction between the adatoms and the magnetic substrates. However, the author's experiments on magnetic Co and non-magnetic Rh atoms on the magnetic substrate do not show much difference in the atoms' diffusion behavior. So, I think this is a specific property of the substrate (Mn/Re(0001)) by itself.

Both is correct, the hexagonal surface symmetry is (electronically) broken due to the row-wise magnetic state AND the magnetic interactions between adatoms and surface play a decisive role. Co and Rh atoms move in a one-dimensional fashion along the $\uparrow\uparrow$ -rows, but due to different mechanisms. In contrast, our DFT calculations for Cu atoms predict two-dimensional anisotropic diffusion disfavoring motion along the $\uparrow\uparrow$ -rows. These results for Cu atoms show that the surface alone cannot determine diffusion directions. We have realized that in the previous version of the manuscript the competition between different processes, i.e., the role of the substrate and the impact of the adsorbate-substrate interaction, were not addressed clear enough. We have rephrased parts of the manuscript to explain these effects, which includes the theory section, where we now show the results for Cu atoms explicitly. We have also added the spin-resolved local density of states for all three

adatoms which shows their different spin-dependent hybridization with the Mn surface atoms (new Supplementary Figure 3).

2) The authors have used voltage pulses to move the atoms. However, I believe that the authors need to verify whether this motion is activated by threshold voltage effect or electric field to distinguish if this motion is activated by specific motional frequency or deformation of potential landscape.

The activation process itself is certainly an interesting issue. We have in fact narrowed down the threshold voltages of single site hopping for Co and Rh atoms to energy windows of 8-50 mV and 200-300 mV, respectively. Doing this more precisely would require extensive measurements and might turn out to be fruitless due to the tip-dependence we have observed so far.

The issue of the electric field is now discussed in the supplement and we find that the electric field has no observable effect on the adatom motion. Since adatom motion can be activated at rather low voltage ($U \sim 10$ mV) and for both voltage polarities (both field directions), it is unlikely that the electric field plays an important factor for the activation process.

We thus have answered the two questions to an extent which is appropriate within our study, as our main focus is the adatom movement and the effect of magnetism thereon.

3) Throughout the manuscript, the authors often discussed spin-flip and change of magnetic moment of adatoms. Regarding this, have the authors performed any tunneling spectroscopy on the adatoms such as dI/dV in order to see any inelastic signal associated with spin/magnetic moment flip? In my opinion, their STM system is around 4.2K so that this measurement may be difficult due to energy resolution. However, imaging the adatoms via SP-tip should be possible to see any change of magnetic moment before/after the atoms being diffused. Have the authors tried this approach?

The reviewer is correct that the energy resolution is likely not sufficient, and together with the expected strong interaction with the electron bath of the metallic substrate one cannot obtain spin-flip excitation spectra for this system. Moreover, dI/dU or d^2I/dU^2 spectroscopy would be difficult for this system, because the Co adatom starts jumping away from the tip already at $U \sim 10$ mV.

Imaging different spin states of adatoms is certainly possible with SP-STM. We have shown this for Co adatoms on fcc-Mn/Re(0001) in new Ref. 12, where the adatom's spin state gives a strong spin-signal in atom manipulation data. In our submitted work we deliberately use tips which show no sign of a spin-polarized current, to exclude additional complications such as spin-torques or magnetic interactions between tip and adatom. Instead of these two standard approaches we have employed a *new* technique, the "kicking", and it turns out, that the "kicked" atom is very sensitive to the magnetic state of the surface.

4) In pg 5, the authors mentioned that "Rh is in the same chemical group and thus iso-electronic to Co, but almost 75% heavier and typically non-magnetic on metal surfaces". I would like to address that the electronic and magnetic properties of the atoms could be completely different for even the same atom depending on which surfaces and which binding site the atom is adsorbed. So, I am uncertain that the authors could claim "iso-electronic", "non-magnetic" and also why the atomic mass matters here. I believed that the electronic and magnetic interactions between the adatom and the substrate is much stronger than mass-related interactions.

The cited sentence is an objective description of the differences between Rh and Co. The term "iso-electronic" is commonly used for elements with the same electronic shell filling. As the valence electrons dominate the hybridization with the surrounding it is expected that iso-electronic adatoms are more similar compared to arbitrary elements. The reviewer is correct that the adsorption site is expected to play an important role in the hybridization as well.

Co often exhibits a magnetic moment when adsorbed on a surface, while Rh is not magnetic as a bulk system and typically not magnetic as a single adatom, unless polarized by a magnetic surface. The distinction between magnetic and "non-magnetic" adatoms is furthermore justified by the calculated magnetic moments displayed in Fig. 5.

We agree with the reviewer that the atomic mass should play no decisive role for resting adatoms, but it might be relevant for adatom dynamics, as we point out in the discussion section. In the revised version of the manuscript we also mention our additional experiments with another iso-electronic adatom, namely Ir, which, however, turns out to be not very mobile.

5) In pg 6-7 and Fig. 5, the authors compared the value of the magnetic moment and energy barrier of adatoms on the substrate. Could you explain how to relate/convert the value of the magnetic moment to energy unit(eV)? This will make the readers understand better that this diffusion behavior is associated with magnetic effect.

Unfortunately, there is no general way to disentangle all relevant effects and to convert magnetic moments to energy barriers. Qualitatively, the energy for the Co adatom rises, when its moment is lowered, which is a consequence of intra-atomic exchange. For Rh atoms the opposite behavior is found, and its energy is decreased when the magnetic moment is lower. Based on the spin-resolved local density of states (LDOS, see new Supplementary Figure 3), we attribute this effect to the spin-dependent hybridization with the surface Mn atoms. As shown by the comparison to the Co and Cu adatom case, the LDOS of the adatom plays an important role for the spin-dependent hybridization at the different saddle points and thereby for the energy barriers. However, quantitative conclusions cannot be drawn from the LDOS.

6) In pg 8, the authors mentioned "Cu" atoms. I wonder if the authors meant Co atoms. Because of the reasons listed above, I am not convinced to recommend this manuscript to be published in Nature Communications.

To make a clearer and more general case, we now discuss the Cu atom calculations in comparison to those for Co and Rh directly in the new Fig. 5 and have added Supplementary Figures 2 and 3 which show the minimum energy paths and spin-resolved LDOS of all adatoms. We have revised the manuscript at different places to further improve readability and hope that the reviewer can be convinced by our additional explanations.

Reviewer #3 (Remarks to the Author):

In this work, F. Zahner and co-workers observed the one-dimensional movement of both magnetic Co atoms and non-magnetic Rh atoms on the surface of AFM ordered Mn layer. Without the inclusion of first-principle calculations, this observation seems to contradict the conclusion that 'magnetism can play a decisive role in controlling atom movement', which is drawn at the end of the article. Even though first-principle calculations show the one-dimensional motion of Co and Rh atoms could result from different kinds of interactions with the magnetic substrate, this serves as weak evidence. Although the conclusion is not surprising, I found the evidence insufficient and the discussion unpersuasive. Therefore, I cannot recommend the publication of this work in Nature Communications. Below are specific issues that need to be addressed:

We thank the reviewer for her/his time and efforts. The reviewer's assessment is based on a misunderstanding of our experiments, and we apologize for not being clear enough in the first version of our manuscript.

1. If the author observed the one-dimensional movement of magnetic Co atoms and free movement of non-magnetic Rh atoms, it directly indicates the decisive role of magnetism in controlling atom

movement. However, the observation of the one-dimensional movement of both magnetic Co atoms and non-magnetic Rh atoms makes the role of magnetism questionable.

The role of magnetism in the adatom movement is not restricted to the magnetic properties of the adatoms, but it is also governed by the magnetic properties of the surface, which is anisotropic in our case. The smoking gun experiments to prove that "magnetism can play a decisive role in controlling atom movement" are shown in Fig. 3 and 4. The fact that both adatom species move in the same direction on a given magnetic domain is insubstantial for that proof. It is surprising, but only at first glance, that "non-magnetic" adatoms such as Rh and Cu also show anisotropic motion. Though this behavior caused some confusion, it demonstrates the novelty and scope of our results: even the motion of non-magnetic adatoms is affected by the magnetic state of the surface!

2. A control experiment above the Neel temperature of Mn layer will ultimately resolve the debate. Without additional experiments, a comprehensive discussion to exclude alternative explanation, such as the influence of anisotropic atomic structure shown in Figure 2d and Extended data Figure 3c, is necessary before concluding the decisive role of magnetism.

The observed one-dimensional movement is a result of the broken surface symmetry (broken by the magnetic state) AND the electronic and magnetic interactions between adatom and surface. Additional temperature-dependent experiments are beyond the scope of this investigation and would not be able to distinguish between purely substrate anisotropy and magnetic adatom-surface interactions, because above the Neel temperature, both effects are zero and adatom motion would be quasi-isotropic. Our view is that the experiments shown in Fig. 3 and 4, combined with the DFT-based NEB calculations, are a more direct and elegant way to unambiguously prove the impact of magnetism.

3. Figure 2a-d shows the anisotropic feature in both SP-STM images and normal STM images of the Mn layer, which can be used for the identification of the spin orientation in the Mn layer. However, the description of the figure is insufficient and therefore confusing. For example, it does not explain the origin of stripe pattern in the normal STM image (Figure 2d). It does not explain what kind of native defects it is and how they correlate with the spin orientation. It seems based on the symmetry. The discussion should be elaborated to ensure that readers have a clear understanding of these images. One suggestion for improvement is to present the simulated SP-STM images and normal STM images of the Mn layer in Figure 2. It demonstrates that the different stripe patterns are observed in SP-STM images and normal STM images, arising from AFM magnetism and structural relaxation, respectively.

We thank the reviewer for her/his advice. We have revised Fig. 2 accordingly and believe that the figure is much clearer and more comprehensible now.

4. The author use blue and yellow color in Figure 2a, b, d to indicate the spin rows. But the blue color is difficult to discern in the STM images. Please consider changing the color for better visibility.

We have moved the line pattern in the new Fig. 2a to an antiphase position for better visibility.

5. When discussing the mobility of Co atoms on fcc-Mn/Re(0001), the descriptions, such as 'Co adatoms on fcc-Mn/Re(0001) are easily moved during standard STM imaging...' and 'The Co atoms are surprisingly mobile on fcc-Mn/Re(0001)', are misleading. This implies the author can distinguish the fcc-Mn/Re(0001) from the STM images and establish the correlation between high mobility of Co atom with the fcc site purely based on experimental observation. But this is not the case. In my opinion, the author should not claim atomic site until they demonstrate the different stability of adatom on fcc and hcp site through first principle calculation.

There seems to be a misunderstanding. fcc-Mn refers to the Mn atoms of the monolayer sitting in fcc positions on the Re(0001) surface. This Mn monolayer stacking has been established experimentally in new Ref. 15. For the Co adatom, we can infer from new Fig. 2e,f that one hollow site is preferred, which

is identified by DFT as the hcp site on the fcc-Mn layer. The new sentence at the beginning of the experimental part should make this point clear.

6. The discussion in section II is a bit difficult to follow. Please consider modify it to improve the clarity.

We have substantially revised the discussion in section II (theory part) as well as Figure 5, now also including results for Cu atoms. We have also added the minimum energy paths of all adatoms (Supplementary Figure 2) as well as their spin-resolved local density of states at the initial states and the saddle points (Supplementary Figure 3) and discuss the role of spin-dependent hybridization between adatom and Mn surface atoms.

7. The conclusion section is too brief. It is not clear how the findings impact 'the related phenomena such as nanostructure growth, molecular self-assembly and catalysis. It needs more elaboration to highlight the significance of the results.

Adatom diffusion is a core mechanism in surface science and we demonstrate for the first time that magnetism can play a decisive role in it. This result is significant in itself and means that all processes which depend on atomic diffusion might be influenced by the magnetic state of the surface as well.

Reviewer #4 (Remarks to the Author):

Zahner and coworkers show a conclusive study of the one-dimensional movement of Co and Re atoms on a Mn thin film grown on Re(0001). STM measurements show a preferred motion along one spatial direction despite the 6-fold symmetry of the non-magnetic lattice. Hence, the authors conclude that the row-wise anti-ferromagnetic structure (AFM) breaks that symmetry and gives a one-dimensional motion. This is most clearly demonstrated by comparing the adatom motion on different rotational domains measured with exactly the same tip apex, thereby excluding many possible tip artifacts. The authors use a comprehensive set of DFT calculations to derive at this conclusion. That theoretical modeling also gives insight into the energy barriers that dominate the adatom motion.

I find these results very clear and the experiments and DFT studies are well executed. I think that it is rather surprising that nobody has studied this effect before, either in SPM or in ensemble techniques. Are the authors sure about the lack of previous studies? I am not aware of any such studies either. I strongly support publication with minor changes.

We thank the reviewer for her/his clear summary and the strong support for publication in Nature Communications.

Important points:

1. Figure 3 makes it clear that there is a one-dimensional motion. However, it is not mentioned what determines upwards vs downwards motion along that axis? This is only answered at the end of the experimental methods. I strongly recommend to move that argument into the main text to not confuse the reader.

We agree and have moved the respective sentence from the methods section to the main text.

2. The threshold for the action spectroscopy is rather unspecific: 8-50mV for Co atoms. In typical action spectroscopy (Kawai group and others), these thresholds are rather precise and typically stem from inelastic excitation of vibrational excitation. Can the authors discuss this comparison in the text?

We believe that there are two aspects which contribute to the difference. First, Kawai and coworkers work with molecules, which have sharp intrinsic vibrational modes in contrast to a single metal atom on a metal surface. Secondly, much of the difference we observe in the threshold voltages for single jumps stems from comparing experiments performed with different micro-tips, presumably as a result

of varying (attractive) forces between tip and adatom. We now refer to the work of the Kawai group in the introduction and the discussion section of the revised manuscript, which includes a brief comparison between molecules and our case.

3. The methods section “B” related to DFT calculations is not clear on the following points:

a. The authors mention that “Films with fcc-stacking of the Mn monolayer were structurally relaxed using the GGA exchange-correlation potential”. Were other functionals used for Co and/or Rh?

We apologize for not being precise here. We used the same GGA exchange-correlation potential for all DFT calculations including the Co, Rh, and Cu adatoms. We have modified the methods section accordingly.

b. The authors mention that they use an “asymmetric film consisting of six Re(0001) layers”. Is the same cell used in the simulation of the STM images? Did changing the code from VASP to FLEUR induce any changes in geometry/force?

We used the structure obtained from our VASP calculations to set up the film calculations performed with FLEUR. In these calculations we also used an asymmetric film.

c. Why was the code changed between the first and last part of the simulations? VASP does have the capability of simulating STM images (for example using the STMpw code). If the change of code was motivated by necessity or ease of performing a certain type of calculation it should be noted here. Authors should specify if the structures had to be re-relaxed or were used as-is.

The FLEUR code was used for the simulation of STM and SP-STM images since it provides an accurate description of the vacuum region. Since the film geometry is implemented in FLEUR the exponential decay of the wave functions and of the charge and magnetization densities is described by the choice of the basis functions in the vacuum region which also exhibit an exponential decay (see e.g. S. Heinze *et al.*, PRB **58**, 16432 (1998)). This allows a more accurate description than in the VASP code which uses plane waves as basis functions in the direction perpendicular to the surface. The FLEUR code also provides powerful tools to analyze the STM and SP-STM images in terms of the spin-dependent electronic structure (see e.g. M. Bode, S. Heinze *et al.*, PRB **66**, 014425 (2002)).

d. It would be helpful to first state all commonalities of the DFT calculations (i.e. cell size, functional, k-points, cutoffs) then contrast the differences for the individual elements.

We have modified the DFT method section to state the commonalities of the DFT calculations for the two codes used first. The differences for the individual elements are stated in their respective places.

Minor points:

4. The title seems a bit unscientific “Kicking atoms”

We have changed the title to "Anisotropic atom motion on a row-wise antiferromagnetic surface".

5. Please add line numbers to make reviewing easier.

We have done that.

6. Page 4 top “In this nonmagnetic data” – this seems like wrong grammar.

We changed the phrase to "When the tunnel current is not spin-polarized,...".

7. Page 7 “latter saddle point” – the word latter seems wrong here

For clarity, we have replaced “latter saddle point” by “the $\uparrow\downarrow$ -bridge site”.

8. Page 10, line 11: the word “respectively” is not matched to anything

For clarity, we added a "for Co and Rh atoms".

Reviewer #5 (Remarks to the Author):

ANSWER TO REVIEWER COMMENTS

Reviewer #1 (Remarks to the Author):

The authors have addressed most of the referees' comments and improved significantly the manuscript. I now recommend its publication.

We thank the reviewer for supporting publication of our manuscript in Nature Communications.

Reviewer #2 (Remarks to the Author):

The authors have addressed the comments and feedback from the first review; however, I am still not convinced that this work is novel enough to be published in Nature Communications. The manuscript may be better suited for more specialized journals.

On the issue of novelty, we fully agree with reviewer #4 who has stated: "I think that it is rather surprising that nobody has studied this effect before, either in SPM or in ensemble techniques. Are the authors sure about the lack of previous studies? I am not aware of any such studies either. I strongly support publication with minor changes."

We have searched the existing literature very carefully and found no experimental work on adatom motion on magnetic surfaces. We encourage the reviewer to substantiate his/her above assessment by citing previous work on the same subject or - in case that is not possible - acknowledge the novelty of our work.

1) The authors claim that this work could be highly useful for material growth, catalysis, and rethinking the growth modes of magnetic films and nanostructures. However, the core of the work involves using an STM tip to move atoms, and while there may be some correlation between atomic motion and magnetic properties, I believe this work is only applicable to very specific cases, not as broadly as the authors suggest. Additionally, I would like to ask the authors to clarify if the Cu, Co, and Ru on the Mn layer atop the Re(0001) system have any particular material properties that should be considered in this context.

- We will discuss in more detail in our answer to the next question that there is a misunderstanding regarding the experimental setup. Based on our results, it is now rather obvious that magnetism can play a key role also for related effects such as nanostructure growth and catalysis.
- On the issue of scope and applicability, we would like to remind the reviewer of a recent work (Ref. 20) on a bilayer of Mn on Ir(111), where the top Mn layer is shifted from the hollow site to the bridge site due to also a row-wise AFM state. It would be very surprising if such a drastic magnetism-induced effect would have no effect on subsequent growth.
- Furthermore, we have not only shown "some correlation" between atomic motion and magnetic properties, but surprisingly that the magnetic state DETERMINES the movement direction of adatoms.
- Concerning the "material properties", the most striking one is the one-dimensional atomic motion on a hexagonal surface, an effect which we demonstrate to be of magnetic origin.

2) The authors repeatedly mention that 'kicking atom' is a new technique and that 'magnetic properties of a surface can play a decisive role in controlling atomic motion'. However, the concept of 'kicking

atoms' has been previously demonstrated in numerous studies under the terms 'lateral and vertical manipulation', performed on metal, semiconductor, and insulating surfaces using electric fields, voltage pulses, Van der Waals forces, etc. There have even been efforts to build magnetic atomic chain structures through atomic manipulation. Furthermore, I still cannot find a clear explanation in the authors' reply regarding how magnetism plays a key role in atomic motion.

For these reasons, I cannot recommend this manuscript for publication in Nature Communications.

- The referee confuses our "kicking" technique with standard atom manipulation techniques. In the latter, the atom position is fully controlled by the tip at all times, typically with the aim to build well-defined atomic arrangements, such as atomic chains or quantum corals. In our case, the adatom motion is initiated with a stationary tip, and neither the direction of motion nor the travel distance is directly controlled with the tip, in contrast to standard atom manipulation techniques. The travel distance can be controlled to some extent by the voltage pulse intensity and length. Because the adatom is not forced by the tip toward a specific direction, the atom motion is very sensitive to the surface potential landscape, and in our case even to the row-wise antiferromagnetic state.
- In addition, we point out in the manuscript, that we often observe single-site jumps of adatoms which are further away from the STM tip during current injection. These initiated single-site jumps are similar to (free) thermally driven jumps and display the same movement direction dictated by the row-wise AFM state, which supports the claim of general applicability.

Reviewer #3 (Remarks to the Author):

I find the manuscript has improved greatly, and most of my comments have been adequately addressed. However, one major concern remains that need to be clarified before the manuscript can be considered for publication.

Geometric Anisotropy versus Magnetic Interactions:

In line 252-253, the authors mentioned that the adatoms are expected to be guided by the geometric anisotropy on surfaces with low structural symmetry, such as Ag(110) or Cu(211). A similar geometric anisotropy is present on Mn/Re(0001), as demonstrated by the prominent stripe pattern in the non-magnetic STM image shown in Figure 2c. This raises the possibility that the geometric anisotropy could also account for the observation of the one-dimensional movement of Co, Rh, and Ir atoms. Before excluding this possibility, it is challenging to conclusively determine the critical role of magnetic interaction in the atomic diffusion.

To be precise, the stripe pattern in Fig. 2c shows an electronic asymmetry of the hexagonal Mn surface layer; in contrast to Ref. 20, we have no direct experimental proof of the lateral shift. Furthermore, in contrast to buckled surfaces like fcc(110) or fcc(211), here the surface Mn layer remains hexagonal (with all nuclei on the same plane) while the symmetry breaking is caused only by the subsurface Re layer.

Nevertheless, the reviewer raises an interesting question which we now can answer at least for the case of Rh adatoms. We have performed additional nudged elastic band (NEB) calculations via DFT for a Rh atom on a Mn layer, which has been forced to remain in the ideal fcc hollow site, i.e. neglecting the magnetically-driven lateral shift. In this unshifted case of the Mn layer, the barriers along $\uparrow\uparrow$ - and $\uparrow\downarrow$ - rows are almost identical, i.e. the potential landscape becomes more symmetrical. Thus, for Rh atoms, the magnetically driven 15 pm lateral shift of the hexagonal Mn layer is in fact needed to explain the one-dimensional adatom motion. We show this new result in the Supplemental Material and added the following sentences on page 8:

"Note, that for the Rh atom motion the magnetism-induced lateral shift observed for the Mn layer on Re(0001) plays a key role. NEB calculations for an unshifted Mn layer lead to almost isotropic minimum energy paths (see Supplementary Figure 5)"

For Co adatoms additional calculations on an unshifted Mn layer did not converge to an unambiguous minimum energy path. We attribute these problems in the NEB calculation to the complex energy landscape due to the interplay of spin and spatial degrees of freedom in a highly symmetric geometrical configuration. Here, our current understanding is that the one-dimensional motion of Co atoms stems from avoiding repeated spin-flips during movement, which favors the $\uparrow\uparrow$ direction.

To make these points clearer, we have modified the abstract and added the term "magnetism-induced" to the term "lateral shift" on page 9.

I believe the authors recognize that the predicted distinct diffusion behaviour of Cu adatoms is key to substantiating their conclusion. Unfortunately, the lack of experimental evidence leaves the diffusion behaviour of Cu atoms remains at a theoretical hypothesis stage. I strongly suggest that the authors include a direct investigation of Cu diffusion on Mn/Re(0001) surface.

Presently, we have no means to prepare Cu atoms on Mn/Re(0001). More importantly, experimental results for Cu adatoms would not affect our conclusions, but would instead only verify or falsify our theoretical prediction for Cu. Experimentally, we have investigated three different adatom species, and the respective data sets shown in Fig. 3 and 4 establish two independent smoking gun experiments for the impact of magnetism onto adatom motion. Even though the 15 pm lateral shift of the hexagonal Mn layer seems to be needed to explain the Rh motion by our DFT calculations, this shift is caused by this particular magnetic state and therefore the magnetism is the key component to explain and understand the observed behavior. Since the case of Cu – though very interesting – might be a distraction from our key results, we have removed a reference to Cu atoms from the abstract.

Reviewer #4 (Remarks to the Author):

The authors properly addressed my concerns. I am fine to publish as is.

We thank the reviewer again for supporting our manuscript from the start.

Reviewer #5 (Remarks to the Author):

Review of “Kicking Co and Rh atoms on a row-wise antiferromagnet”

F. Zahner et al., have reported STM and DFT studies on lateral motion of Co and Rh atoms induced by voltage pulses from a STM tip on magnetic surfaces. The authors have measured and simulated that the atoms' preferred movement is associated with magnetic properties of the surfaces. I believe that the manuscript is in general written well. However, I do not think that this work meets the high standard of novelty required by Nature Communications for the following reasons.

1) In the introduction section (pg 2-3), the authors mentioned that “For bulk systems, however, experiments indicate that the magnetic states can affect diffusion” and “On magnetic surfaces, so far no diffusion experiments have been performed”. This sounds to me that the diffusion direction of the adatoms could be controlled by the magnetic interaction between the adatoms and the magnetic substrates. However, the author's experiments on magnetic Co and non-magnetic Rh atoms on the magnetic substrate do not show much difference in the atoms' diffusion behavior. So, I think this is a specific property of the substrate (Mn/Re(0001)) by itself.

2) The authors have used voltage pulses to move the atoms. However, I believe that the authors need to verify whether this motion is activated by threshold voltage effect or electric field to distinguish if this motion is activated by specific motional frequency or deformation of potential landscape.

3) Throughout the manuscript, the authors often discussed spin-flip and change of magnetic moment of adatoms. Regarding this, have the authors performed any tunneling spectroscopy on the adatoms such as dI/dV in order to see any inelastic signal associated with spin/magnetic moment flip? In my opinion, their STM system is around 4.2K so that this measurement may be difficult due to energy resolution. However, imaging the adatoms via SP-tip should be possible to see any change of magnetic moment before/after the atoms being diffused. Have the authors tried this approach?

4) In pg 5, the authors mentioned that “Rh is in the same chemical group and thus iso-electronic to Co, but almost 75% heavier and typically non-magnetic on metal surfaces”. I would like to address that the electronic and magnetic properties of the atoms could be completely different for even the same atom depending on which surfaces and which binding site the atom is adsorbed. So, I am uncertain that the authors could claim “iso-electronic”, “non-magnetic” and also why the atomic mass matters here. I believed that the electronic and magnetic interactions between the adatom and the substrate is much stronger than mass-related interactions.

5) In pg 6-7 and Fig. 5, the authors compared the value of the magnetic moment and energy barrier of adatoms on the substrate. Could you explain how to relate/convert the value of the magnetic moment to energy unit(eV)? This will make the readers understand better that this diffusion behavior is associated with magnetic effect.

6) In pg 8, the authors mentioned “Cu” atoms. I wonder if the authors meant Co atoms.

Because of the reasons listed above, I am not convinced to recommend this manuscript to be published in *Nature Communications*.